# Perfluoroalkyl Substances (PFASs) in Rivers and Drinking Waters from Qingdao, China

**DOI:** 10.3390/ijerph19095722

**Published:** 2022-05-08

**Authors:** Guohui Lu, Pengwei Shao, Yu Zheng, Yongliang Yang, Nan Gai

**Affiliations:** 1Key Laboratory of Eco-Geochemistry, Ministry of Natural Resources of China, National Research Center for Geo-Analysis (NRCGA), Beijing 100037, China; luguohui@mail.cgs.gov.cn (G.L.); pengwei-shao@outlook.com (P.S.); yuzheng520@outlook.com (Y.Z.); ylyang2003@163.com (Y.Y.); 2Qingdao Junray Intelligent Instrument Co., Ltd., Qingdao 266109, China; 3Health Care Security Administration of Shizhong District, Zaozhuang 277000, China

**Keywords:** perfluoroalkyl substances, surface water, drinking water sources

## Abstract

Perfluoroalkyl substances (PFASs) in rivers; drinking water sources (reservoirs and groundwater); and various types of drinking waters (tap waters, barreled pure waters, and bottled mineral waters) in Qingdao, Eastern China were quantified by high-performance liquid chromatography tandem mass spectrometry (HPLC-MS/MS). The total concentrations of PFASs (ΣPFASs) in the river waters ranged from 28.3 to 292.2 ng/L, averaging 108 ± 70.7 ng/L. PFBS was the most abundant compound, with a maximum concentration of 256.8 ng/L, followed by PFOA (maximum concentration: 72.4 ng/L) and PFBA (maximum concentration: 41.6 ng/L). High levels of PFASs were found in rivers in the suburban and rural areas. The estimated annual mass loading of the total PFASs to Jiaozhou Bay (JZB) was 5.9 tons. The PFASs in the drinking water reservoirs were relatively low. The ΣPFASs in the tap water ranged from 20.5 ng/L to 29.9 ng/L. Differences in the PFAS levels and composition profiles were found among barreled water at different market sites and for different brands of mineral water products. The sequence of the contamination levels of the waters related to drinking water was reservoir water > tap water > barrel water > groundwater > bottled mineral water. The PFASs in drinking water may not pose a serious risk to the drinking water consumers of Qingdao City.

## 1. Introduction

Perfluoroalkyl substances (PFASs) are widely used in the industry, agriculture, and people’s daily lives because of their excellent hydrophobic and oleophobic characteristics due to the strong polarity and high bonding energy of the C–F bond in the molecules. They are difficult to degrade and easy to enrich in organisms [1]. PFASs have been widely detected in aquatic environments [2,3], drinking waters [4,5], wildlife, and the human body [6,7]. PFASs with saturated carbon–fluorine bonds appear to be incompletely removed during conventional chemical/physical (coagulation, flocculation, sedimentation, and filtration) and biological (activated sludge) treatment processes [8].

Perfluorooctanoate (PFOA) and perfluorooctane sulfonate (PFOS) were reported as the main compounds in the early publications on PFASs in tap water in China [4,5]. PFOA and PFOS are considered to have carcinogenic potency in animals [9]. In 2009, PFOS was classified as a persistent organic pollutant (POP) under the Stockholm Convention. The early reports showed that PFOS was the predominant type of PFAS present in Chinese tap water collected from several large cities such as Shenzhen, Shanghai, and Hong Kong during 2004–2008, accounting for at least 50% of the total PFASs [5]. In a recent survey of PFASs in tap water from 17 cities in China, Tan et al. (2017) observed a lower proportion of PFOS in tap water (averagely accounting for 6.4% of the total PFAS). They found that PFOA became the predominant type of PFAS and increasing proportions of perfluoro-n-butanoic acid (PFBA) in areas north of the Yangtze River and sodium perfluoro-1-hexaneslufonate (PFHxS) in areas south of the Yangtze River [10]. The status of the contamination level of PFASs in tap water has become a pressing public concern in recent years [11]. Information about the relations between surface waters and tap water with respect to the contamination level and composition profile of PFASs is still lacking, especially under the context of replacements of PFOA and PFOS with compounds less hazardous but relatively more resistant to degradation, which has been implemented in China and worldwide since 2009. Reports on the occurrence of PFASs in bottled pure water and mineral water are sparse [12]. Such information may help evaluate the regional trends of PFAS contamination levels and composition profiles and reveal the PFAS sources for tap waters and the removing efficiency for PFASs in various types of commercial drinking water production.

Qingdao, a large coastal city in North China, may represent an interesting case study to investigate the occurrence and exposure of PFASs along the rural–urban environmental and socioeconomic gradients. The tap water in Qingdao City is a mixed water from several raw water sources. In this study, the contamination level of the PFASs in river water; drinking water sources (reservoirs and groundwater); and in various types of drinking water (tap water, barreled pure water, and bottled mineral water) from Qingdao were investigated. The objective was to assess the impact of PFAS contamination in surface water on drinking water and human health. The mass loading of PFASs in the rivers and human risk via the route of the drinking water were also discussed.

## 2. Materials and Methods

### 2.1. Sample Collection

Field sample collections of surface water, groundwater, tap water, barreled pure water, and bottled mineral water were completed in March 2017. Twenty-four river water and two reservoir water samples were collected. Three shallow groundwater sampling sites (6–10 m deep) in Xiashan Village of Chengyang District, Lijiaxia Village of Laoshan District, and Beizhai Village of Laoshan District were selected in this study. Yukuang Reservoir and Laoshan Reservoir in Laoshan District were selected as the surface water source sites for drinking water. Seven tap water and six barreled pure water samples were collected in six districts of Qingdao City. The barreled pure water was sampled in areas with dense populations. Three locally popular brands of mineral water produced in Qingdao were purchased in the market. All sampling sites are shown in Figure 1.

The surface water samples were taken from 0–0.2 m below the water surface using a stainless-steel bucket, then transferred into 2L polypropylene (PP) bottles. Groundwater samples were directly collected from the hand-pressed wells using 2L PP bottles. Samples were immediately transported to the laboratory and stored at 4 °C until analyzed. Detailed information about each sampling site is summarized in Appendix A for the surface water and drinking water samples, respectively, in the Appendix A.

### 2.2. Materials and Reagents

Perfluorooctane sulfonamide (PFOSA); a mixture of PFBA, perfluoro-n-pentanoic acid (PFPeA), perfluoro-n-hexanoic acid (PFHxA), perfluoro-n-heptanoic acid (PFHpA), PFOA, perfluoro-n-nonanoic acid (PFNA), perfluoro-n-decanoic acid (PFDA), perfluoro-n-undecanoic acid (PFUnDA), perfluoro-n-dodecanoic acid (PFDoDA), perfluoro-n-tetradecanoic acid (PFTrDA), perfluoro-n-tetradecanoic acid (PFTeDA), perfluoro-n-hexadecanoic acid (PFHxDA), perfluoro-n-octadecanoic acid (PFOcDA), potassium perfluoro-1-butanesulfonate (PFBS), PFHxS, PFOS, and sodium perfluoro-1-decanesulfonate (PFDS); and mixture of ^13^C_4_-PFOS, ^13^C_2_-PFDA, ^13^C_5_-PFNA, ^13^C_4_-PFOA, ^13^C_4_-PFBA, ^13^C_2_PFHxA, ^13^C_2_PFUnDA, ^13^C_2_PFDoDA, and ^18^O_2_PFHxS were purchased from Wellington Laboratories Inc. (Guelph, Ontario, Canada). Purities of the individual chemicals and mass-labeled chemicals are >98%. Oasis weak anion exchange solid-phase extraction cartridges (WAX; 6cc, 150 mg) were purchased from Waters Corp. (Milford, MA, USA). Methanol (LC-MS grade) was purchased from Merck Ltd. (Darmstadt, Germany), and ammonium solution (25%) was purchased from Sigma-Aldrich GmbH (Seelze, Germany). Milli-Q water was used during the experiment.

### 2.3. Sample Pretreatment

The water samples were extracted using solid-phase extraction (SPE) with WAX cartridges based on the method of ISO25101 [13]. In brief, 4 mL of 0.1% ammonia/methanol, 4 mL of methanol, and 4 mL of Milli-Q water passed through the cartridges successively. The unfiltered water samples with mass-labeled standards spiked were passed through the cartridges at the rate of 1 to 2 drop/s. After loading all samples, the cartridges were rinsed with 4 mL of 25 mM acetate buffer solution (pH 4), and the PFASs were eluted with 4 mL of methanol followed by 4 mL 0.1% NH_4_OH. The effluent was concentrated to 1 mL by nitrogen.

### 2.4. Instrumental Analysis

Separation of the PFASs was performed using a tandem mass spectrometer (API 4000, Applied Biosystems Inc., Framingham, MA, USA) coupled with high-performance liquid chromatography (HPLC) (Agilent Technologies 1200, Santa Clara, CA, USA). The LC column used was an RSpak JJ-50 2D ion exchange column (2.0 mm × 150 mm, 5 μm; Shodex, Tokyo, Japan). The injection volume was 10 μL. The analytes were eluted with a 50 mmol/L ammonium acetate–methanol mixed solution (volume ratio 2:8) in isocratic elution mode at a flow rate of 300 μL min^−1^ for 20 min. The column temperature was 40 °C. The electrospray ionization voltage was 4000 V using a negative mode ion source, and the ion source temperature was 350 °C. The curtain gas pressure was 69 KPa, the ion source GAS1 pressure was 344.7 KPa, the ion source GAS2 pressure was 344.7 KPa, and the collision gas pressure was 34.5 KPa. The target compounds were detected using multiple reaction monitoring (MRM) in negative ion electrospray mode. Calibration curves for the instrument were prepared with a series of seven concentrations at 2, 10, 50, 200, 1000, 5000, and 25,000 pg/mL. The instrumental response of the target analytes was confirmed for quantification using individual chromatograms.

### 2.5. Quality Assurance and Quality Control (QA/QC)

To achieve lower detection limits, all of the accessible polytetrafluoroethylene (PTFE) and fluoropolymer materials in the HPLC instrument and apparatus were replaced with materials made of polyetheretherketones (PEEK). The procedure and travel blanks for the water were collected. A recovery test was carried out using both mass-labeled and native standard chemicals. Matrix recoveries were conducted by adding mass-labeled mixed standard surrogates and native standard chemicals to real samples.

Quantitative responses according to the amount of standards added were evaluated. Concentrations of analytes were calculated using an external calibration curve. The data for blanks and recoveries were obtained in duplicate for every 12 samples to ensure stable repeatability. Samples should be reanalyzed if the results of the blanks and recovery exceeded the acceptable range. When repeatability was achieved and the signal/noise ratio (S/N) was ≥10, the lowest concentration of the target analyte was defined as the limit of quantification (LOQs) of the method. Appendix A in the Supplementary Information presents the LOQs and recoveries for individual PFASs in the water samples from this study. For matrix-spiked water samples, only PFDS, PFHxDA, and PFOcDA showed recoveries less than 80%.

## 3. Results and Discussion

### 3.1. Occurrence of PFASs in River Waters

#### 3.1.1. Detection Rates and Contamination Levels

The concentrations of individual PFAS compounds at each sampling site in this study are provided in Appendix A. Table 1 presents the statistics on the analytical results for the PFASs in the water samples from the six rivers, the drinking water sources, and drinking waters in Qingdao. Of the eighteen PFASs monitored in the river waters, twelve were detected, including eight perfluorinated carboxylic acids (PFCAs), three perfluoroalkyl sulfonates (PFSAs), and PFOSA. The detection rates of nine compounds (PFBA, PFPeA, PFHxA, PFHpA, PFOA, PFDA, PFOS, PFHxS, and PFBS) were 100%. The detection rates of PFNA, PFUnDA, and PFOSA were 96.3%, 66.7%, and 33.3%, respectively. Most of the targeted long-chain (C10–C18) PFASs (i.e., PFDoDA, PFTrDA, PFTeDA, PFHxDA, PFOcDA, and PFDS) were not detected in all the samples.

The total concentrations of the PFASs (ΣPFASs) in the river water samples of Qingdao ranged from 28.3 to 292.2 ng/L, averaging 108 ± 70.7 ng/L. The lowest ΣPFASs occurred at the sampling site W9 in Licun River, while the highest were at site W6 in Zhangcun River. The compound with the maximum concentration was PFBS (256.8 ng/L), followed by PFOA (68.4 ng/L) and PFBA (41.6 ng/L). Compared with the reported ΣPFASs in surface waters in other large cities in China, the highest ΣPFASs observed in this study were higher than those in the surface waters of Beijing (2.9–222.6 ng/L, target PFASs, including 11 PFCAs, 3 PFSAs, and 3 chlorinated polyfluoroalkyl ether sulfonic acids) [14]; Shanghai (39–212 ng/L, target PFASs, including 11 PFCAs, 5 PFSAs, and PFOSA) [15]; and Hangzhou (94.3–179.3 ng/L, target PFASs, including 13 PFCAs and 4 PFSAs) [16] but lower than those in Nanchang (146.2–586.2 ng/L, target PFASs, including 13 PFCAs and 4 PFSAs) [17]; Changshu (15.6–480.9 ng/L, target PFASs, including 13 PFCAs and 4 PFSAs) [18]; and Taihu Lake in Eastern China (164 to 299 ng/L, target PFASs, including 5 PFCAs and 2 PFSAs) [19].

#### 3.1.2. Spatial Distributions

Qingdao is a highly unevenly distributed city with respect to the population, industry, and agriculture. The Moshui River, the Hongjiang River, the Baisha River, and the upstream of the Licun River flow through the rural areas of Qingdao, while the Haibo River and Zhangcun River and the downstream of the Licun River flow through the urban areas of Qingdao. In recent decades, most of industries in Qingdao have been moved from the urban areas to the rural areas. Higher contamination levels of PFASs, as well as special composition profiles of PFASs from packaging and industrial parks, were observed in the rural areas of Qingdao (Figure 2). The differences in the concentrations and compositions of PFASs from the rural area to the urban area are particularly obvious for Licun River. The average ΣPFASs were in the sequence of Hongjiang River > Moshui River > Zhangcun River > Haibo River > Baisha River > Licun River > reservoirs. Due to tides, the seawater flowing backward into the lower reaches of Haibo River, Licun River, and Moshui River may dilute the PFAS concentrations at the river mouth sampling sites (W3, W20, and W24).

A higher proportion of PFBS in ΣPFASs was observed in Haibo River and Zhangcun River (averaging 35.6% and 42.5% of the total PFASs, respectively). The proportion of PFBS in the ΣPFASs decreased from the upper reaches to the lower reaches of Haibo River and Zhangcun River but increased from the upper reach to the lower reach in Licun River. Though short-chain compounds have been widely used as the substitute of PFOS and PFOA in this part of China [20], the degradation of long-chained perfluoroalkane sulfonates can also be an important source of PFBS [21,22]. Several waste water treatment plants (WWTPs) are located near sites W2, W19, and W25. It has been reported that PFBS exists in effluents from many WWTPs in Europe and, recently, in China [23,24,25].

The general contamination level of PFASs in Moshui River and Hongjiang River (ΣPFASs: 186.1 and 223.4 ng/L, respectively) was more serious than in the other rivers, characterized by relatively higher proportions of PFOA and PFOS. In contrast, high levels of the short-chain compounds PFBS (256.8 ng/L) and PFHxS (33.8 ng/L) were observed in Zhangcun River at sites W6 and W7, respectively. Elevated PFHxS concentrations were observed when compared to earlier reports in the Yangtze River Delta region in South China [26]. PFHxS has already been listed as a substance of very high concern by the European Chemicals Agency (Helsinki, Finland) [27]. However, the production and use of PFOS and PFHxS still take place in China for firefighting foam [28,29]. There is a firefighting training site and a furniture factory in the vicinity of site W6 and several electronic manufactures near site W7. PFHxS can be used as a repellent and is widely utilized in firefighting foam, printing inks, and sealants. It is not expected to undergo hydrolysis or photolysis, and no biodegradation is expected [30]. Large proportions of PFOS (14.9% at W19 and a mean of 15.0% at W21 and W22) and PFOSA (12.1%) in the total PFASs were observed in Moshui River and Hongjiang River. The concentrations of PFOSA were also high (44.0 and 13.8 ng/L at W22 and W21, respectively). Along these two rivers, poultry breeding and packaging have developed, which may relate to PFOSA, as it is used in food packaging [1]. PFOSA is an important PFOS precursor and was phased out by 3 M in the United States during 2000–2002, but it has grown in China by other producers. On the other hand, in Licun River and Baisha River, PFOA, PFBA, and PFHxA were the three most abundant compounds. Higher PFHxA concentrations were also found in Moshui River and Hongjiang River, which flow through the rural areas of Qingdao. PFHxA is regarded as a highly stable and ultimate transformation product from several precursors [31].

#### 3.1.3. Mass Loading of PFASs to the Sea

The rivers in this study eventually flow into Jiaozhou Bay, and the coastal sea water and sediments will become the sink of the PFASs, which may have a direct impact on the water quality and ecology of Jiaozhou Bay. Therefore, the annual input of the PFASs from the five rivers (Zhangcun River merges with Licun River before flowing into the sea) into the coastal waters was estimated as a first-order approximation based on the PFAS concentrations and the water discharge data available. The annual input of the PFASs from the riverine outlets into the sea in Qingdao are shown in Table 2. The estimation results show that a total of 5.9 tons per year of 10 PFAS compounds were discharged into Jiaozhou Bay, with individual PFAS loads ranging from 0.038 to 1.32 tons per year. It should be noted that these data were obtained based on the concentrations of PFASs in their dissolved forms and in the dry season of the region. The annual inputs of PFOA, PFBS, PFBA, and PFOS into the sea were 1.3, 1.2, 0.9, and 0.4 tons per year, respectively. These estimated discharges of the PFASs were much lower than those of the large rivers in China and the world, such as the Yangtze River, which contributed a considerable proportion (>60%) of riverine discharge for PFASs in China [26]. For example, the annual loads of PFOA, PFOS, and ΣPFASs in the Yangtze River were 6.8, 8.2, and 20.7 tons, respectively [32]; the annual loads of PFBA, PFBS, and ΣPFASs from the river Rhine into the North Sea were 10.5, 5.1, and 17 tons, respectively [33]. The mass loading of the PFASs into the rivers of Qingdao was slightly higher than that in Daling River in Northeast China, with annual mass loadings of PFBA, PFBS, and ΣPFASs into the Bohai Sea of 0.14, 0.23, and 0.46 tons, respectively [34]. The annual discharges of PFBA, PFHxA, PFOA, PFOS, and the total PFASs in Jiulong River in Southeastern China were 0.05, 0.16, 0.04, 0.09, and 0.47 tons, respectively [35].

The mass load contributions of all the rivers into Jiaozhou Bay were in the sequence of Moshui River > Baisha River > Licun River > Haibo River > Hongjiang River. Moshui River, Baisha River, and Licun River contributed about 86% of the total PFAS loading, which may be attributed to the higher volumes of water discharge of the three rivers. Moshui River constituted the largest proportion (36.6%), followed by Baisha River (33.6%). Wang et al. (2016) estimated the total Chinese riverine mass discharges of PFOA (mean: 80.9 t/y; range: 16.8–168 t/y), which were in good agreement with the theoretical PFOA emission estimates (range: 17.3–203 t/y), whereas the riverine had mass discharges of PFOS (mean: 3.6 t/y; range: 1.9–5.6 t/y) [36]. Our results indicated that the PFOA input from the rivers in Qingdao contributed to approximately less than one percent of China’s total PFOA emissions.

### 3.2. Occurrence of PFASs in Drinking Water Sources

#### 3.2.1. Reservoirs

Since the municipal water supply in Qingdao takes the way of multiple-source water mixing, the surface waters of Laoshan Reservoir and Yukuang Reservoir were collected as the representatives of large and small reservoirs, respectively. Though there are several reservoirs as the drinking water sources for Qingdao City, some rural villagers in Laoshan District and Chengyang District of Qingdao City still take the groundwater as their drinking water source; therefore, the reservoirs and groundwater samples are compared and discussed together. Comparisons between PFASs in the raw waters for drinking purposes and in the tap water of Qingdao City will be discussed in the next section.

The ΣPFASs in the surface waters of Laoshan Reservoir (W13) and Yukuang Reservoir (W8) were 39.4 ng/L and 28.6 ng/L, respectively, which were much lower than those in the rivers. However, the concentrations of PFBA in the reservoirs were comparable to those of several rivers in Qingdao. PFOA and PFBA were the dominant PFASs in the two reservoirs. With more PFBS and PFHxS but less PFBA and PFOA, the composition profile of the PFASs of Laoshan Reservoir was different from that of Yukuang Reservoir. The presence of PFASs in these reservoirs could be attributed to sewer leakage or a discharge of raw sewage and storm water runoff from the surrounding villages. The concentrations of PFOA were 12.96 and 7.48 ng/L for Laoshan Reservoir and Yukuang Reservoir, respectively. These values were higher than those of Guanting Reservoir (PFOA ranging from 0.55 ng/L to 2.3 ng/L) [37] and Miyun Reservoir (ΣPFASs ranging from 5.30 ng/L to 8.50 ng/L) [38], both in Beijing, China, but comparable to Marina Reservoir, Singapore (PFOA ranging from 8 ng/L to 37 ng/L) [39].

#### 3.2.2. Groundwater

Detailed analytical results for the groundwater and drinking water samples are shown in Table 1 and Appendix A. The ΣPFASs in the groundwater samples (GW1–GW3) ranged from 0.41 to 19.5 ng/L, averaging 8.3 ± 10.0 ng/L. The detection rates of PFBS and PFBA were 100%. Overall, PFBA and PFBS were the two compounds with the highest concentrations found in the three groundwater samples, accounting for 35.1% and 25.7% of the average ΣPFASs, respectively, reflecting the downward movement of highly water soluble PFASs such as PFBS [22]. The highest ΣPFASs were observed at GW1, located along Zhangcun River and surrounded by many wood, furniture, and wood carving factories. The most abundant PFASs at GW1 were PFBA, PFOS, PFBS, and PFOA, accounting for 26.9% 24.8%, 22.7%, and 17.0% of the ΣPFASs, respectively. Although the contamination level at this site was higher than those of the other two groundwater sampling sites, on the whole, the ΣPFASs in this groundwater sample were less than those of the reservoir waters (average: 24.4 ng/L) and the tap waters (average: 24.5 ng/L; see the following section on tap water). The lowest ΣPFASs in groundwater were observed at GW3 situated in Laoshan Mountain characterized by a high terrain with no industrial zone around it and a relatively small population. Only PFBA, PFHpA, and PFBS were detected at GW3, and their contents were low. Sampling site GW2 is also close to the Laoshan Scenic Spot at the foot of the mountain, but the surrounding residential areas are larger, with a denser population and an animal breeding industry. The ΣPFASs in the groundwater in Qingdao are comparable to the published values for groundwater in the cities in Eastern China (0.20–8.5 ng/L) [40] and Northeastern China (3.1 ng/L) [3].

### 3.3. Occurrence of PFASs in Drinking Water

#### 3.3.1. Tap Water

Nine target compounds (PFOA, PFNA, PFBA, PFPeA, PFHxA, PFHpA, PFOS, PFHxS, and PFBS) in the tap water samples (TW1–TW7) from six districts of Qingdao City were detected in all of the samples, with detection rates of 100%. The ΣPFASs ranged from 20.5 ng/L to 29.9 ng/L, averaging 24.5 ± 3.6 ng/L. PFOA and PFBA were the most abundant PFAS compounds in the tap water samples. The proportion of each compound in the total measured PFASs was as follows: PFOA (31.6%) > PFBA (21.3%) > PFHxS (14.2%) > PFOS (6.9%) = PFPeA (6.9%) > PFBS (6.8%) > PFHpA (5.4%) = PFHxA (5.4%) > PFNA (1.5%). In contrast, PFHxA and PFOA were the two major PFASs in the tap water of most cities in Eastern China [17]. The Yellow River, Baisha River, and Laoshan Reservoir are the three most important drinking water sources for Qingdao City. The Yellow River is the second-largest river in North China. In the last decade, a project “Yellow River Diversion to Qingdao” has been implemented to bring water through canals to Qingdao so as to meet the increasing needs of fresh water in Qingdao City. A previous study showed that PFBA and PFOA were the two major PFAS compounds in the Yellow River water [41].

The average concentration of PFOA in the tap water samples from Qingdao was 7.8 ± 1.5 ng/L and that of PFBA was 5.2 ± 0.4 ng/L. PFHxS was the third-highest compound, ranging from 1.9 ng/L to 4.7 ng/L and averaging 3.5 ± 1.1 ng/L. The other PFAS compounds were generally below 2 ng/L, with detection rates less than 7%. The replacement of PFOS with less hazardous compounds has been implemented in China since 2009. Compared with the early reports on PFASs in tap water in China [4,5], the proportion of PFOS in the tap water samples of Qingdao was lower (averagely accounting for 6.9% of the total PFASs), and the average concentration was 1.7 ± 0.5 ng/L. In contrast, PFOS was once the predominant type of PFAS present in Chinese tap water (mean: 3.9 ng/L) more than 10 years ago during 2004–2008 [5]. The PFASs in tap water from Qingdao were at a level from low to moderate amounts of contamination, higher than those of Tokyo, Japan (0.72–95 ng/L, target PFASs, including 11 PFCAs and 5 PFSAs) [42]; Thailand (0.58–1.15 ng/L, target PFASs, including 8 PFCAs and 3 PFSAs) [43]; and Turkey (0.08–11.27 ng/L, target PFASs, including 7 PFCAs and 3 PFSAs) [12].

It was noted that the detection rates, the concentrations, and the proportions of each PFAS compound were similar in all tap water samples in Qingdao City (Figure 2). This situation is not accidental. There are three major tap water plants in Qingdao, i.e., Xianjiazhai, Baishahe, and Laoshan Waterworks, which deal with the water of the Yellow River, the Dagu River, and the reservoirs, respectively. The three major waterworks mix the treated tap waters evenly in the Qingdao Urban Pipelines and then enter the residents’ water pipelines so that the daily tap water used by the residents is the mixed water of each waterwork. Therefore, the similarity of the characteristics of the tap water in each district is very high, which was preliminarily verified by this study.

#### 3.3.2. Commercial Drinking Waters

In addition to tap water, residents in Qingdao also consume commercial barreled pure water and bottled mineral water, especially in the workplace. The concentrations of PFASs in the barreled pure water (BW1–BW6) and bottled mineral water (MW1–MW3) samples are presented in Table 1 and Appendix A. The ΣPFASs in the barreled pure water ranged from 0.2 ng/L to 28.4 ng/L, with a median of 0.9 ng/L and mean: 8.8 ± 12.9 ng/L. The PFASs in most of the barreled pure waters were low, except for the two barreled pure water samples BW4 and BW5 (ΣPFASs were 28.4 and 22.2 ng/L, respectively). Compared with the tap waters and groundwater, BW1, BW2, BW3, and BW6 were obviously much better with respect to the PFAS contents. It has been reported that the PFAS contents of bottled waters were generally lower than the tap waters in The Netherlands, Greece, and Turkey [12,44].

It can be seen that there is a great difference between different barreled pure waters. In BW1 and BW2, and PFBA was the only PFAS compound detected. PFBA and PFOA were detected in BW3 and BW6, though at very low levels (both below 0.5 ng/L). In contrast, the ΣPFASs in BW4 and BW5 were as high as those of the tap waters (averaging 24.5 ng/L), and similar composition profiles to those of the tap waters were observed, with nine compounds (PFOA, PFNA, PFBA, PFPeA, PFHxA, PFHpA, PFOS, PFHxS, and PFBS) being detected, suggesting that they were unqualified pure water products, if not fakes.

The mineral water samples were from two different manufacturers. MW1 and MW2 were from Laoshan Mineral Water Company, Qingdao, and MW3 from Kelan Mineral Water Company, Qingdao. Laoshan Mineral Water with a high mineral content and low salinity is from the wells penetrating the cracks of deep granite of 117 m underground at the Laoshan Scenic Spot area. The production of Laoshan Mineral Water is less than 20,000 tons a year due to too-few deep-well water sources. In contrast, Kelan Mineral Water is an ordinary mineral water that is produced from relatively shallower wells in Taiping Mountain, Qingdao. The price of Kelan Mineral Water is lower than that of Laoshan Mineral Water.

The analytical results of the bottled mineral water samples showed that there were differences in the PFAS composition profiles among the different brands of mineral water products. Only one compound (PFBA) was detected in MW1 and MW2, with a very low concentration of 0.22 ng/L in both samples. In contrast, PFBA, PFPeA, PFHxA, and PFBS were detected in MW3, with a ΣPFASs of 7.02 ng/L. The concentration of PFBA (4.28 ng/L) in MW3 was about 20 times higher than that in MW1 and MW2 but was only one-third of that in the tap water samples, indicating that the mineral water produced from the relatively shallower aquifer was still cleaner than the tap waters with respect to PFAS contamination. 

Table 1 summarized the statistics on PFASs in drinking water source sites and various types of drinking waters in Qingdao. The order of the contamination levels expressed by the sequence of the ΣPFASs from high to low is reservoir water (34.0 ng/L) > tap water (24.5 ng/L) > barreled pure water (8.8 ng/L) > groundwater (8.3 ng/L) > bottled mineral water (2.5 ng/L). The USEPA has developed lifetime drinking water health advisories for PFOS [45] and PFOA [46] of 70 ng/L for each and 70 ng/L for the sum of the concentrations of PFOS and PFOA when both occur in the same drinking water supply. Based on this, the potential risk for the drinking water consumers in Qingdao is generally low. The Chinese standards for drinking water quality [47] which will be implemented in 1 April 2023, set the limit values of the PFOA and PFOS concentration in drinking water as 80 ng/L and 40 ng/L. Based on this, the potential risk for the drinking water consumers in Qingdao is generally low.

## 4. Conclusions

PFASs were widely detected at relatively high levels in river waters in Qingdao. Compared with the rivers of other large cities in China and the world, the ΣPFASs in the river waters in Qingdao were at moderate to high contamination levels. 

The rivers flowing through the rural areas of Qingdao showed high PFAS contamination levels, because most of the industries in Qingdao have been moved from the urban areas to the rural areas in recent decades. 

PFOA, PFBA, and PFBS were the predominant PFAS compounds in the raw waters (reservoir water and groundwater) for drinking purposes in Qingdao, while PFBA, PFOA, and PFHxS were the dominant components in the tap waters. The contamination level of the PFASs in the reservoir waters was higher than in the tap waters, but the contamination level of the PFASs in the groundwater was lower than in the tap waters. The PFASs in drinking water may not pose a serious risk to the drinking water consumers of Qingdao City.

## Figures and Tables

**Figure 1 ijerph-19-05722-f001:**
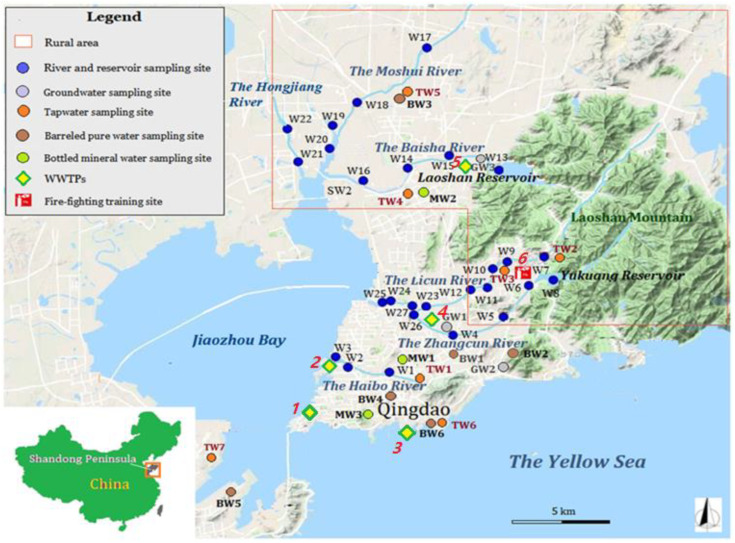
Map of the water sampling sites in Qingdao. The numbers in red italics denote the locations of potential anthropogenic contamination sources: (1) Qingdao Tuandao Sewage Treatment Plant. (2) Qingdao Haibo River Water Operation Co., Ltd. (Qingdao, China). (3) Qingdao Maidao Sewage Treatment Plant. (4) Qingdao Licun River Sewage Treatment Plant. (5) Jimo Sewage Treatment Co., Ltd. (Qingdao, China). (6) Qingdao Fire Training Base.

**Figure 2 ijerph-19-05722-f002:**
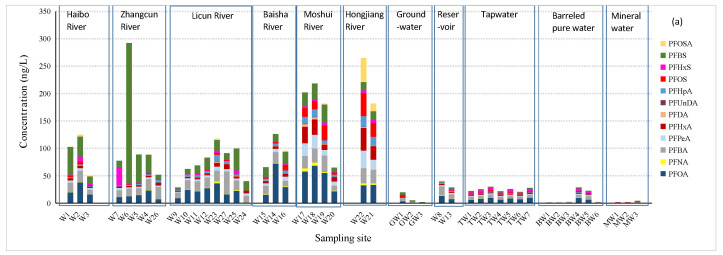
(**a**) Concentrations and (**b**) composition profiles of PFASs in different water matrices in Qingdao.

**Table 1 ijerph-19-05722-t001:** Statistics on the detected PFASs in water samples from the rivers, the drinking water sources, and drinking waters in Qingdao.

Water Type	Statistics	PFBA	PFPeA	PFHxA	PFOA	PFNA	PFDA	PFUnDA	PFHpA	PFBS	PFHxS	PFOS	PFOSA
The Haibo River	Detection rate (%)	100	100	100	100	100	100	0	100	100	100	100	66.7
	Ave. * (ng/L)	15.76	3.91	3.45	24.27	0.36	0.15	<LOQ	2.01	33.63	4.68	2.77	1.70
	SD * (ng/L)	6.36	0.69	0.82	12.07	0.08	0.13	<LOQ	0.60	18.44	4.09	2.02	2.01
	Median * (ng/L)	17.60	3.91	3.79	19.44	0.39	0.21	<LOQ	2.12	36.60	2.36	2.24	1.70
	Min. * (ng/L)	8.68	3.22	2.52	15.36	0.27	<LOQ	<LOQ	1.36	13.88	2.28	1.06	0.28
	Max. * (ng/L)	21.00	4.60	4.04	38.00	0.42	0.24	<LOQ	2.54	50.40	9.40	5.00	3.12
The Zhangcun River	Detection rate (%)	100	100	100	100	100	100	0	100	100	100	100	0
	Ave. (ng/L)	16.18	3.03	3.09	13.74	0.28	0.18	<LOQ	2.76	71.94	7.85	0.61	<LOQ
	SD (ng/L)	4.73	0.61	1.20	6.07	0.27	0.10	<LOQ	1.27	104.80	14.53	0.45	<LOQ
	Median (ng/L)	13.68	2.77	2.84	12.64	0.34	0.17	<LOQ	2.44	32.40	1.34	0.54	<LOQ
	Min. (ng/L)	12.12	2.42	2.16	7.04	<LOQ	<LOQ	<LOQ	1.60	7.88	1.00	<LOQ	<LOQ
	Max. (ng/L)	19.92	3.90	2.91	23.40	0.54	0.29	<LOQ	2.81	256.80	33.84	1.24	<LOQ
The Licun River	Detection rate (%)	100	100	100	87.5	87.5	75.0	25.00	87.5	100	87.5	100	12.5
	Ave. (ng/L)	28.48	14.75	14.86	27.13	10.49	8.85	2.87	13.45	26.39	11.45	12.14	1.49
	SD (ng/L)	9.47	1.67	2.47	11.22	0.99	0.78	0.21	3.46	10.76	1.93	1.11	0.33
	Median (ng/L)	17.76	4.06	3.45	22.10	0.63	0.35	<LOQ	3.85	16.24	1.29	0.93	<LOQ
	Min. (ng/L)	10.00	1.40	1.13	<LOQ	<LOQ	<LOQ	<LOQ	<LOQ	4.64	<LOQ	<LOQ	<LOQ
	Max. (ng/L)	41.60	7.36	8.84	36.40	3.16	2.46	0.60	11.72	40.00	5.80	3.65	0.92
The Baisha River	Detection rate (%)	100	100	100	100	100	100	66.7	100	100	100	100	33.3
	Ave. (ng/L)	15.09	6.20	4.65	38.87	1.12	0.63	0.18	4.50	18.19	0.97	4.67	0.17
	SD (ng/L)	4.55	1.42	0.81	29.90	0.19	0.13	0.16	0.73	4.27	0.38	4.55	0.30
	Median (ng/L)	15.68	6.24	4.40	29.20	1.15	0.58	0.25	4.88	17.40	0.92	2.06	<LOQ
	Min. (ng/L)	10.28	4.76	4.00	15.00	0.92	0.52	<LOQ	3.66	14.36	0.61	2.03	<LOQ
	Max. (ng/L)	19.32	7.60	5.56	72.40	1.30	0.77	0.30	4.96	22.80	1.37	9.92	0.51
The Moshui River	Detection rate (%)	100	100	100	100	100	100	100	100	100	100	100	50
	Ave. (ng/L)	22.04	16.37	18.98	50.75	3.54	2.66	0.55	9.14	23.49	3.29	15.37	0.19
	SD (ng/L)	9.32	8.71	11.63	20.25	2.43	2.16	0.34	4.84	10.84	1.94	9.47	0.23
	Median (ng/L)	24.50	16.54	18.76	56.50	3.54	2.56	0.50	9.60	26.68	3.40	15.38	0.14
	Min. (ng/L)	8.84	7.56	8.20	21.60	1.31	0.69	0.25	3.68	7.88	0.82	3.78	<LOQ
	Max. (ng/L)	30.32	24.84	30.20	68.40	5.76	4.84	0.95	13.68	32.72	5.56	26.96	0.47
The Hongjiang River	Detection rate (%)	100	100	100	100	100	100	100	100	100	100	100	100
	Ave. (ng/L)	50.73	50.17	55.40	55.24	35.58	34.33	33.51	45.03	43.77	36.73	55.85	52.60
	SD (ng/L)	1.61	9.98	11.46	0.54	0.81	0.61	0.04	2.97	0.59	0.14	11.06	21.35
	Median (ng/L)	26.10	25.26	33.10	32.86	3.37	1.49	0.27	17.54	15.66	5.10	33.78	28.90
	Min. (ng/L)	24.96	18.20	25.00	32.48	2.80	1.06	0.24	15.44	15.24	5.00	25.96	13.80
	Max. (ng/L)	27.24	32.32	41.20	33.24	3.94	1.93	0.30	19.64	16.08	5.20	41.60	44.00
Reservoirs	Detection rate (%)	100	100	100	100	100	100	0	100	100	100	100	0
	Ave. (ng/L)	12.58	2.41	1.91	10.22	0.56	0.15	<LOQ	2.18	3.12	0.60	0.29	<LOQ
	SD (ng/L)	5.85	0.03	0.24	3.87	0.26	0.02	<LOQ	0.84	2.27	0.85	0.26	<LOQ
	Median (ng/L)	9.22	1.22	1.07	7.05	0.41	0.09	<LOQ	1.51	2.69	0.72	0.27	<LOQ
	Min. (ng/L)	8.44	2.39	1.74	7.48	0.37	0.14	<LOQ	1.58	1.51	<LOQ	0.10	<LOQ
	Max. (ng/L)	16.72	2.43	2.08	12.96	0.74	0.17	<LOQ	2.77	4.72	1.20	0.47	<LOQ
Groundwater	Detection rate (%)	66.7	66.7	66.7	66.7	33.33	0	0	100	66.7	33.3	66.7	0
	Ave. (ng/L)	18.39	16.80	16.77	17.71	8.41	<LOQ	<LOQ	25.19	18.18	8.45	17.99	<LOQ
	SD (ng/L)	2.64	0.17	0.11	1.72	0.19	<LOQ	<LOQ	0.13	2.21	0.28	2.68	<LOQ
	Median (ng/L)	1.40	0.20	0.19	0.85	<LOQ	<LOQ	<LOQ	0.33	1.52	<LOQ	0.43	<LOQ
	Min. (ng/L)	0.20	<LOQ	<LOQ	<LOQ	<LOQ	<LOQ	<LOQ	0.11	<LOQ	<LOQ	0.00	<LOQ
	Max. (ng/L)	5.26	0.35	0.20	3.32	0.32	<LOQ	<LOQ	0.33	4.44	0.48	4.84	<LOQ
Tap water	Detection rate (%)	100	100	100	100	100	0	0	100	100	100	100	0
	Ave. (ng/L)	5.15	1.69	1.31	7.78	0.38	<LOQ	<LOQ	1.32	1.65	3.54	1.72	<LOQ
	SD (ng/L)	0.36	0.20	0.12	1.51	0.08	<LOQ	<LOQ	0.19	0.39	1.15	0.45	<LOQ
	Median (ng/L)	5.10	1.68	1.25	7.66	0.35	<LOQ	<LOQ	1.32	1.62	4.14	1.71	<LOQ
	Min. (ng/L)	4.62	1.44	1.18	6.04	0.30	<LOQ	<LOQ	1.03	1.08	1.87	1.09	<LOQ
	Max. (ng/L)	5.80	2.04	1.52	9.64	0.52	<LOQ	<LOQ	1.65	2.34	4.74	2.32	<LOQ
Barreled water	Detection rate (%)	100	33.3	33.3	66.7	33.33	0	0	33.3	33.3	33.3	33.3	0
	Ave. (ng/L)	2.21	0.59	0.42	2.70	0.12	<LOQ	<LOQ	0.45	0.55	1.28	0.51	<LOQ
	SD (ng/L)	2.77	0.92	0.65	4.11	0.20	<LOQ	<LOQ	0.72	0.86	2.07	0.84	<LOQ
	Median (ng/L)	0.66	<LOQ	<LOQ	0.28	<LOQ	<LOQ	<LOQ	<LOQ	<LOQ	<LOQ	<LOQ	<LOQ
	Min. (ng/L)	0.18	<LOQ	<LOQ	<LOQ	<LOQ	<LOQ	<LOQ	<LOQ	<LOQ	<LOQ	<LOQ	<LOQ
	Max. (ng/L)	6.32	1.91	1.36	9.52	0.48	<LOQ	<LOQ	1.64	1.89	4.76	1.95	<LOQ
Mineral water	Detection rate (%)	100	33.3	100	0	0	0	0	33.3	33.3	0	0	0
	Ave. (ng/L)	1.57	0.59	0.23	<LOQ	<LOQ	<LOQ	<LOQ	0.05	0.13	<LOQ	<LOQ	<LOQ
	SD (ng/L)	2.34	1.03	0.30	<LOQ	<LOQ	<LOQ	<LOQ	0.09	0.22	<LOQ	<LOQ	<LOQ
	Median (ng/L)	0.22	0.00	0.06	<LOQ	<LOQ	<LOQ	<LOQ	<LOQ	<LOQ	<LOQ	<LOQ	<LOQ
	Min. (ng/L)	0.22	0.00	0.05	<LOQ	<LOQ	<LOQ	<LOQ	<LOQ	<LOQ	<LOQ	<LOQ	<LOQ
	Max. (ng/L)	4.28	1.78	0.58	<LOQ	<LOQ	<LOQ	<LOQ	0.16	0.38	<LOQ	<LOQ	<LOQ

* Ave.: Average concentration; SD: Standard deviation; Min.: Minimum concentration; Max.: Maximum concentration.

**Table 2 ijerph-19-05722-t002:** The annual input of the PFASs from the riverine outlets to the sea in Qingdao.

	Watershed Area	Annual Water Discharge	PFOA	PFNA	PFBA	PFHxA	PFDA
	km^2^	104 m^3^/a	kg/a	kg/a	kg/a	kg/a	kg/a
The Haibo River	14	381	39.4	0.7	22.3	6.5	0.0
The Licun River	132	3576	2.2	0.0	321.9	90.9	0.9
The Baisha River	215	3133	616.4	24.2	217.0	117.4	16.2
The Moshui River	317	3734	543.4	32.9	222.4	206.3	17.4
The Hongjiang River	56	558	125.0	10.5	93.8	94.0	4.0
		PFUnDA	PFHpA	PFOS	PFHxS	PFOSA	Total
		kg/a	kg/a	kg/a	kg/a	kg/a	kg/a
The Haibo River		0.0	3.5	2.7	6.1	0.7	126
The Licun River		0.0	0.0	18.3	1.1	0.0	960
The Baisha River		5.3	103.0	209.4	28.9	10.8	1990
The Moshui River		6.3	92.7	95.0	20.5	543.4	2169
The Hongjiang River		0.9	58.1	97.6	18.8	51.9	683

## Data Availability

Not applicable.

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
