# Peer review of "Perfluoroalkyl Substances (PFASs) in Rivers and Drinking Waters from Qingdao, China"

_ijerph, 2022, doi:10.3390/ijerph19095722_

Round 1

Reviewer 1 Report

The manuscript “IJERPH-1667801” aim to define the concentration of perfluoroalkyl substances (PFASs) in rivers, drinking water source sites (reservoirs and 11 groundwater), and various types of drinking waters (tap waters, barreled pure waters, and bottled 12 mineral waters) in Qingdao, Eastern China, in order to improve the knowledge about their distribution.

The paper appear well-structured and clear. Few improvements are needed (see below)

I believe the manuscript should be published  after minor revision.

Comments (P = page#/R = row#):

  • Figure 1.

I suggest to insert the location of potential anthropogenic contamination sources.

  • Introduction/ Results and discussion sections

I suggest to insert information about treatment technologies employing to remove these contaminant considering also other constituents of  waters (e.g. essential constituent for human heath) . Authors can take inspiration from the following works:

  • Banks, D., Jun, B. M., Heo, J., Her, N., Park, C. M., & Yoon, Y. (2020). Selected advanced water treatment technologies for perfluoroalkyl and polyfluoroalkyl substances: A review. Separation and Purification Technology, 231, 115929.

  • Fuoco, I., Apollaro, C., Criscuoli, A., De Rosa, R., Velizarov, S., & Figoli, A. (2021). Fluoride Polluted Groundwaters in Calabria Region (Southern Italy): Natural Source and Remediation. Water, 13(12), 1626.

  • Results and discussion sections (P6,R252)

Are there maximum permissible limits for these substances in waters established by national or international agencies?

The World Health Organization (WHO) has set parameters about it?

The authors should report these information and if available the authors have to compare it with their detected data. 

  • Figure 3.

In other words, do the data represent the concentrations of PFAS in the waters of the relevant reservoirs before and after the treatment? If this interpretation is true, please explain it better in the text to make it clearer.

  • References

Add the new references in the references list.

Reviewer 2 Report

The article entitled "Perfluoroalkyl substances (PFASs) in rivers, drinking water 2 source sites, and various drinking waters of Qingdao, China" is an interesting report on the analysis of concentrations of PFAS in waters of China using mostly HPLC-MS/MS. It requires minor revision and minor English review. 

Line 74 - Delete sample collections. These are not described in this section. Section seems to be too lengthy. Mention Figure 1 if appropriate.

Lines 213-218 - Good point. How do ensure that these results are comparable? Please elucidate this, but keep the relevant discussion about other studies measuring same chemicals.

Line 228 - Can you give examples of polluting industries?

LIne 241 - "PFBS is relatively more resistant to 241 degradation, either by photolysis and biodegradation [22,23" Why is this relevant? Elucidate the reader.

Line 261 - It is not clear  how poultry breeding relates to food packaging

Lines 377-381 See lines 213-218

Reviewer 3 Report

Please see the comments below.

Reviewer 4 Report

This paper presents a very interesting subject that is within the aims and scope of International Journal of Environmental Research and Public Health. Congratulations to the Authors. My suggestion is acceptance with considering the following modifications/revisions.

ABSTRACT

Abstract should be started with a general problem statement about the PAFSs instead of the specific objectives.

Line 16: Please give full name of PFBS, PFOA, and PFBA.

Line 18: What do you mean by “drinking water source site waters”? Please revise.

Line 19: Please give full name of PFHxS.

Please end this section with some implications of the findings in a broader context. For example, what can others learn from your investigation? How can they apply your findings to their own case studies?

KEYWPRDS

Please do not repeat the words used in Title again.

INTRODUCTION

Line 44: “carcinogenic potential” or “carcinogenic potency”?

Lines 70-72: I couldn’t understand this statement. Please revise.

MATERIALS AND METHODS

Line 89: Please give full name of WWTP.

Line 93: “is” has been duplicated. Please revise.

Lines 115-124: Which protocol/guideline did you use to analyze the samples?

RESULTS AND DISCUSSION

Line 201: Add “s” to “PFAS”

Line 246: Please revise the unit as “ng/L”.

Table 1: What is the difference between “0” and “-”?

Conclusions

OK

Round 2

Reviewer 3 Report

Authors have made changes accordingly. MS can be accepted.

Author Response

      Dear Reviewer:

      This time several changes in the manuscript have been made.

1. The title of the manuscript has been changed as “Perfluoroalkyl substances (PFASs) in rivers and drinking waters from Qingdao, China”.

2. Unified the expression of some specific nouns, for example:“tapwater”change to “tap water”, “highperformance”change to “high-performance”; “eastern China ”and “southeastern China” change to “the eastern China” and “the southeastern China”.

3. The forms of some verbs are corrected. For example in part 3.1.2, “which may relatesthe PFOSA” change to “which may relate the PFOSA”.

4. The 41st reference was listed repeatedly, and the serial number of the reference is corrected.

 We would like to express our sincere thanks to you for the constructive and positive comments.

Reviewer 4 Report

My suggestion is acceptance.

Author Response

Dear Reviewer:

This time several changes in the manuscript have been made.

1. The title of the manuscript has been changed as “Perfluoroalkyl substances (PFASs) in rivers and drinking waters from Qingdao, China”.

2. Unified the expression of some specific nouns, for example:“tapwater”change to “tap water”, “highperformance”change to “high-performance”; “eastern China ”and “southeastern China” change to “the eastern China” and “the southeastern China”.

3. The forms of some verbs are corrected. For example in part 3.1.2, “which may relatesthe PFOSA” change to “which may relate the PFOSA”.

4. The 41st reference was listed repeatedly, and the serial number of the reference is corrected.

 We would like to express our sincere thanks to you for the constructive and positive comments.

This manuscript is a resubmission of an earlier submission. The following is a list of the peer review reports and author responses from that submission.